# The Use of Molecular Dynamics Simulation Method to Quantitatively Evaluate the Affinity between HBV Antigen T Cell Epitope Peptides and HLA-A Molecules

**DOI:** 10.3390/ijms23094629

**Published:** 2022-04-22

**Authors:** Xueyin Mei, Xingyu Li, Chen Zhao, Anna Liu, Yan Ding, Chuanlai Shen, Jian Li

**Affiliations:** 1Key Laboratory of Developmental Genes and Human Disease, Ministry of Education, School of Life Science and Technology, Southeast University, Nanjing 210096, China; 220203770@seu.edu.cn (X.M.); xingyu0077@outlook.com (X.L.); 220193506@seu.edu.cn (A.L.); 2Department of Microbiology and Immunology, Medical School, Southeast University, Nanjing 210009, China; zhaochen418107@163.com (C.Z.); ding_y2015@163.com (Y.D.)

**Keywords:** hepatitis B virus, affinity, MM-GBSA, residue scanning, molecular dynamics

## Abstract

Chronic hepatitis B virus (HBV), a potentially life-threatening liver disease, makes people vulnerable to serious diseases such as cancer. T lymphocytes play a crucial role in clearing HBV virus, while the pathway depends on the strong binding of T cell epitope peptide and HLA. However, the experimental identification of HLA-restricted HBV antigenic peptides is extremely time-consuming. In this study, we provide a novel prediction strategy based on structure to assess the affinity between the HBV antigenic peptide and HLA molecule. We used residue scanning, peptide docking and molecular dynamics methods to obtain the molecular docking model of HBV peptide and HLA, and then adopted the MM-GBSA method to calculate the binding affinity of the HBV peptide–HLA complex. Overall, we collected 59 structures of HLA-A from Protein Data Bank, and finally obtained 352 numerical affinity results to figure out the optimal bind choice between the HLA-A molecules and 45 HBV T cell epitope peptides. The results were highly consistent with the qualitative affinity level determined by the competitive peptide binding assay, which confirmed that our affinity prediction process based on an HLA structure is accurate and also proved that the homologous modeling strategy for HLA-A molecules in this study was reliable. Hence, our work highlights an effective way by which to predict and screen for HLA-peptide binding that would improve the treatment of HBV infection.

## 1. Introduction

Infection with Hepatitis B virus (HBV) is a common cause of liver cancer and other diseases and can even progress to death due to its high carcinogenicity. HBV has now been identified as a type of DNA virus with a complete envelope that solely targets human and orangutan cells [1]. Studies have revealed that HBV cannot lead to liver cells lesions by itself and the damage of liver cells is mainly induced by the associated immune response [2]. When livers are infected with HBV, HBV can induce the immune response of cytotoxic T cells and antibodies, which protects people from disease. During this process, the formation of the peptide–MHC complex is a prerequisite for T cell recognition. Studies have shown that HBV peptides with a strong binding affinity are presented by MHC molecules to T cells [3]. Therefore, an analysis of the change in the affinity between HBV polypeptides and MHC molecules proves helpful in identifying HLA-restricted binding peptides of HBV.

Major Histocompatibility Complex (MHC) encodes different products specialized to various species of mammals and is also called Human leukocyte antigen (HLA) in humans. HLA class I molecule is a kind of heterodimer formed by heavy chain α and β2-microglobulin and is expressed on white blood cells, and plays an important role in the immune system [4]. According to the characteristics of HLA-coding genes, the HLA class I gene can be divided into the following two categories: classical class I genes (HLA-A, -B, -C), which are highly genetically polymorphic, and non-classical I genes (HLA-E, -F, -G). Classical Class I genes are responsible for delivering the peptide (typically 8–12 amino acids in length) to CD8+ T cells, while HLA class II genes (HLA-DR, HLA-DQ, and HLA-DP) encode HLA proteins presenting peptides with a length of 13–17 amino acids to CD4^+^ T cells. There are also HLA III gene regions, which contain genes encoding complement components C2, C3 and C4 and B factors, etc. An increasing number of studies have shown that HLA alleles are associated with certain diseases, and HLA-A are known to be related to immune response pathways in different cancers, among which HLA-A * 33 is associated with susceptibility to HBV infection [5]. However, the distribution of these HLA alleles may vary significantly by region and nationality. For the same pathogen, different populations may have the same overall disease mechanism, but the heterogeneity of HLA alleles in each population may exhibit different immune patterns against the pathogen. The relationship between HLA alleles and HBV infection in Chinese and Northeast Asian populations remains unclear [6]. In addition, there are many studies on the correlation between HLA-II genotyping and HBV infection and limited studies regarding HLA-I. Therefore, in this study, we focus on 13 HLA-A genotypes with a total frequency of approximately 95% in Chinese and Northeast Asian populations to explore the association between HBV infection and T-cell immune responses based on the HLA structure which determines the function. Studies have demonstrated that the antigen binding grooves of chain α are highly polymorphic in HLA molecules from diverse species, which presents different peptide libraries [7]. However, the main chain conformation of peptides, with a bulge in the middle and both ends embedded in a groove, is conserved and proved to be favorable for HLA binding and T cell recognition [8]. The higher the conformational complementarity between the antigen peptide and the peptide binding groove of the HLA molecule, the stronger the affinity between the two.

Traditional biological experiments used to verify the binding of HBV peptides and HLA molecules are highly demanding, such as enzyme-linked immunosorbent assay (ELISA), flow cytometry, and other methods [9,10]. They require strict inclusion criteria to determine the research object and to collect effective peripheral blood collection from the patients with hepatitis B infection [11]. Moreover, it is also difficult to rank the affinity results between different HLA alleles based on the basis of experimental phenotypic results alone. Therefore, in order to overcome the shortcomings of traditional experiments, we implemented a structure-based prediction method for peptide–HLA binding affinity, which has been shown to be more suitable for the prediction of all types of MHC receptors [12].

To date, a number of tools and methods based on sequence and structure have been developed to predict antigen-specific HLA-restricted peptides. Here, we classify sequence-based methods into five major categories, namely (i) a motif search-based approach including SYFPEITHI [13]; (ii) artificial neural networks (ANNs), including NetMHC [14], NetMHCpan [15]; (iii) a support vector machine (SVM) including SVMH [16], (iv) hidden Markov models (HMMs) and (v) quantitative matrices-driven methods (QMs), including BIMAS [17], EPIJEN [18]. Nevertheless, their sensitivity and specificity are uncertain due to conformational changes and selection between peptides and HLA molecules, resulting in some false positive results. In addition, studies have shown that the flexibility of peptides can also increase the difficulty of protein docking [19], which presents great challenge to structure prediction. Therefore, in order to improve the accuracy of prediction, understanding the conformational changes of HLA molecules and peptides is of vital importance. As is well known, the higher the conformational complementarity between the antigen peptide and the peptide binding groove of the HLA molecule, the stronger the affinity between two [20]. Reasonable binding conformation provides the basis for studying the affinity of HLA-HBV antigenic peptides. However, some individual HLA genotypes do not have crystal structures, and even if individual HLA genotypes have several crystal structures in PDB, their bound peptides are significantly different from HBV peptides, which makes it difficult to study the interaction between HLA and HBV peptides. In this study, we used two approaches to obtain a reasonable initial conformation of binding for the assessment of affinity. As for HLA genes, which have PDB, we analyzed all structure-bound peptide sequences and selected the HLA structure with the most similar properties to HBV peptides as candidate structures used for amino acid scanning mutation analysis. For HLA genes with no structure, we constructed an HLA heterodimer using homology modeling and molecular docking, so as to obtain a reasonable initial peptide binding conformation for subsequent analysis. Then, the binding peptides in the candidate HLA structures were mutated into HBV peptides, respectively, by means of residue scanning, and the binding free energy was calculated by MM-GBSA to represent changes in affinity and stability. Considering the complexity of peptide and HLA binding, some structures need to be sampled by molecular docking and molecular dynamics simulations to analyze the protein–protein interaction. In general, we examined the binding affinity between 45 HBV antigenic peptides and HLA and obtained 352 affinity values. According to the test results, the affinity of HBV peptides and HLA genotypes can be evaluated and ranked, which complements the traditional experimental methods. At the same time, this research process also provides convenience for the prediction and screening of HLA molecular binding peptides, and generates new insights for the prevention and clinical diagnosis of liver cancer.

## 2. Results

### 2.1. Selection of HLA Structures for Affinity Analysis

In this Section, structures with a different peptide-bond conformation of HLA obtained from PDB were screened. Studies have shown that the P2 and PΩ residues of antigenic peptides play an important role in the combination of antigenic peptides to HLA molecules, so the HLA structures with similar P2 and PΩ residues with those of HBV peptides were reserved. For example, in the process of analyzing the affinity between the HBV peptide (Sequence: FLWEWASVR) and HLA-A * 02:01, HLA-A * 33:03 and HLA-A * 24:02, we used two different approaches to obtain the candidate HLA structures used for subsequent mutation analysis. As for HLA-A * 33:03, we modelled the structure and evaluated its quality, for which all parameters are shown in Appendix A. As for HLA-A * 02:01 and HLA-A * 24:02, all structures that were collected were analyzed. As Table 1 shows, the P2 anchor residue of the HBV-positive peptide FLWEWASVR is lysine (Lys). As for HLA-A * 02:01, which has the greatest number of structures available in PDB, we chose 1AO7, 1BD2, 1DUZ, 1QEW, 1JHT, 3I6G, 5ENW, 5F9J and 5FA3 (PDB ID) for HLA-A * 02:01, which have the same anchor sites. As for HLA-A * 24:02, its available structures were relatively scare, so two structures with PDB ID of 3I6L and 2BCK were selected according to the resolution and peptide length.

### 2.2. Acquisition of the Initial Conformation of HBV Peptide-HLA Docking

Some HLA genotypes do not have a crystal structure, so we used homology modeling and molecular docking to construct an HLA heterodimer. However, modelled structures do not have the original bond peptide. Considering the high similarity between the modelled structure and the template structure of the antigen binding groove, we used an online tool, CoDockPP, to dock the predicted HLA heterodimer with the peptide ligand derived from its template [21]. After that, the predicted complex conformations obtained were comprehensively sorted according to the score value and ligand RMSD, and the top 10 conformations were selected for subsequent analysis. Here, we selected the top-ranking conformation after the docking of four model structures (Figure 1). It can be seen that the conformation of the template ligand and the predicted ligand almost overlap [22]. The ligand RMSD (Å) values presented in Table 2 are almost all less than 1 Å, indicating that the possibility of predicting structural changes is small, and that it can be used for subsequent amino acid mutation tests [23].

### 2.3. Analysis of Binding between HBV Peptides and HLA Molecules

We used the Residue Scanning function of Schrödinger software (2020-1, LLC, New York, NY, USA) to manually mutate HLA-bound peptides into HBV positive peptides, and used the MM-GBSA method to calculate the affinity of the mutated peptides. Detailed information of the residue scanning results is shown in Appendix A. We can find that different HBV positive peptides have significantly different affinity with HLA-A. According to the value of ΔAffinity, the most likely structure of HLA-A molecules can be screened preliminarily and the quantitative ranking of different HLA-A molecules can be made based on the prediction results. In this study, we selected parts of the ligands presenting negative affinity, identified from the results of some HLA mutations (Table 3). For the HBV-positive peptides, FLWEWASVR and ETVLEYLVSV, both have a strong affinity to HLA-A * 02:01. Different HBV-positive peptides have different limitations, such as VWLSVIWMMW, which has a strong interaction with HLA-A * 24:02. For the above three HBV positive peptides, we also selected the complex with the strongest affinity to the HLA structure to produce a 2D map (Figure 2). In order to clearly show the change of the interaction caused by HBV peptides compared with HLA original bond peptides, we calculated the number of different types of interactions (Table 4).

### 2.4. Dynamics Simulation

A disulfide bond is a type of covalent bond, easily formed between two cysteine residues, which has a vital influence on the stability of protein space and protein activity. In the process of predicting the structural affinity between an HBV positive peptide and HLA molecule, the polypeptide chain of some HLA molecules contains two cysteines, which is linked by disulfide bonds, and the software generally chooses to skip them, as is the case for the polypeptide chain sequence SSSSCPLSK from one of the structures of HLA-A * 11:01. For this type of structure, we adopt peptide docking and optimize the structure through molecular dynamics simulation to obtain RMSD results so as to determine whether the protein ligand docking is stable (Figure 3). The stable region was selected to calculate the binding free energy of the complex by MM-GBSA method. It can be used to compare the binding affinity between different HLA genotypes and HBV-positive peptides. For example, we performed a 100 ns molecular dynamics simulation for a complex composed of the positive HBV peptide SMYSCCCTK and HLA-A * 11:01\A * 02:01\A * 03:01 and another complex composed of HBV peptide ETVLEYLVSV and HLA-A * 26:01\ A * 11:01\ A * 02:01. At the initial stage of the simulation, the system is in a relatively violent state of motion and the distance between each atom has not yet found an equilibrium point. As the simulation progresses, in around the last 50 ns interval, the RMSD value of the docking structure tends to be flat. According to the corresponding number of trajectory frames, the average binding free energy can be obtained (Table 5). When the value of binding free energy is smaller, the binding force between proteins is stronger. Our study can sequence the binding conditions of an HBV positive peptide and HLA molecule. HBV positive peptide SMYPSCCCTK binds to HLA-A * 02:01 best, with a binding free energy of −80.997 kcal/mol, followed by HLA-A * 03:01, while HLA-A * 11:01 has the worst binding free energy. Similarly, the binding condition of the positive peptide ETVLEYLVSV to HLA molecule A * 11:01 was the best, was in the middle for A * 26:01 and was found to be the worst for A * 02:01.

### 2.5. Binding Analysis of 45 HBV Epitopes and HLA-A Alleles

By analyzing the affinity results of all the epitopes with HLA-A alleles, we finally demonstrated the binding advantages of these epitopes with different HLA in Table 6. It can be found that most of the epitopes have a strong interaction with HLA-A * 24:02, HLA-A * 02:01 and HLA-A * 11:01. Finally, we identified 19 epitopes that strongly interacted with different HLA alleles and were more likely to be present on the cell surface, which was 68% consistent with our experimental verification [24]. Among them, ten epitopes (ILCWGELMNL, SYVNVNMGL, WFHISCLTF, VWLSVIWMMW, MMWYWGPSL, LYSILSPFL, RLKVFVLGG, LYSSTVPCF, LYSSTVPVF and FYPKVTKYL) have a high binding affinity to HLA-A * 24:01 and eight epitopes (FLPSDFFPSI, WFHISCLTF, ETVLEYLVSV, ILSTLPETTV, SPISSIFSR, SMYPSCCCTK, FLWEWASVR and MMWYWGPSL) have a high binding affinity to HLA-A * 02:01. Epitopes STLPETTVVR and QAGFFLLTR are more likely to bind to HLA-A * 11:01, while epitopes CPGYRWMCLR and FLWEWASVR have a high binding affinity to HLA-A * 33:03 and epitope FLPSDFFPSI has a high ability binding to HLA-A * 02:07. The affinity values of all epitopes with HLA-A alleles can be found in Appendix A.

## 3. Discussion

The identification of immunogenic peptides is a multi-step process that requires a great deal of time and labor to complete. For antigens with known sequences, the traditional experimental method synthesizes a large number of overlapping peptides and then screens the peptides with immunogenicity using cell experiments, which is time-consuming and labor-intensive. In recent years, the rapid development of bioinformatics has made it faster and more accurate to predict mutations and to obtain high-resolution HLA typing from tumor genomes. Croft et al. studied the immunogenicity of viral antigenic peptides presented by MHC class I molecules on the surface of virus-infected cells and found that peptide–MHC binding affinity was the best predictor of immunogenicity [25]. In this study, the affinity values of different peptide–MHC complexes were predicted using bioinformatic methods, thereby providing a new idea for screening immunogenicity candidate neoantigens and shortening the development time of vaccines.

At present, sequence-based peptide–MHC affinity prediction methods have become the most potent candidate for affinity prediction. However, their dependence on training data makes these methods more effective at predicting high-frequency HLA alleles than low-frequency ones. Therefore, a new, non-offset method is still needed to better predict the affinity of each HLA molecule to the peptide. Based on the following two reasons, we proposed a structure-based prediction method, in which a P-MHC complex model was constructed using a calculation method, providing affinity predictions according to the model. Firstly, although MHC molecules have a high degree of polymorphism, there are only differences in amino acids at individual positions among different molecules. According to the principle that the primary sequence of a protein determines spatial conformation, we can use homology modeling techniques to construct MHC molecules of interest for study. Secondly, the binding pattern of the peptide terminus to MHC class I molecules is highly conservative and is constrained by the F pocket in the antigen binding slot. We can use the conserved peptide terminal to design a docking scheme and obtain a reasonable conformation of a peptide–MHC molecule for molecular dynamics simulations.

In this study, considering the multiple complex conformations of individual HLA alleles, we performed a sequence analysis on the bond peptides inherent in the HLA crystal structure. Since the binding force between the peptide and HLA molecule is mainly provided by the hydrogen bond network formed by the P2 and PΩ sites of the peptide and the antigen binding groove of HLA, in cases where the residues of the main anchor site of the HLA-bound peptide were consistent with the positive HBV peptide, we reserved the HLA structure for subsequent residue scanning analyses to obtain a reasonable peptide–HLA docking conformation. For HLA alleles without an established structure, we used homology modeling, molecular docking and dynamics simulation to obtain a reasonable docking conformation of the HBV peptide and HLA molecule.

Based on molecular dynamics simulations and MM-GBSA calculations, which performed well in predicting the relative binding ability of protein–ligand, we obtained 352 affinity values of 45 HBV positive peptides and 13 HLA genotypes present in different populations. The 45 epitopes here were verified as real epitopes using ex vitro enzyme-linked immunosorbent spot (ELISPOT) and in vitro co-culture (using patients’ peripheral blood mononuclear cells). Previously, we experimentally verified the binding ability of these peptides to HLA by flow cytometry, and categorized affinity levels as low, medium, and high [24]. However, this kind of experimental method has limitation on obtaining the exact value of affinity. Here, our bioinformatic methods can effectively quantitatively predict the situation and provide specific affinity values that can be compared between different groups, which complements the limitations of the experimental method and is suitable for preliminary prediction in the absence of experimental samples or in the case of an excessive sample size so as to provide clues. In summary, our predicted results were highly consistent with the level of qualitative affinity measured by the competitive peptide-binding experiment [24].

However, the molecular dynamics simulation method we adopted, strictly speaking, is a simulation of the model under the thermodynamic system to obtain the state of the model at the simulated time and set temperature, which does not constitute a real experimental system. Moreover, some studies have shown that the temperature of the dynamic simulation will have different effects on the secondary and tertiary structures of proteins. If the temperature is too high, it may be of little referential significance for practical experiments.

The CD8+ T cell response is critical to HBV infection [26]. There are also some databases indicating the presence of HLA class I epitopes in hepatitis B virus, such as Hepitopes, but they are mainly limited to HLA-A * 02:01, A * 24:02 or B * 07:02, which are common supertypes in Caucasians [27]. Therefore, we hope to predict the binding affinity between HLA molecules and HBV antigenic peptides by studying the effects of mutations on binding affinity and stability, and to reasonably screen out the restrictive antigenic peptides targeting high frequency HLA-A in Chinese and Northeast Asian populations. Based on our research process, we can carry out quantitative sequencing according to the affinity value between an antigen peptide and HLA, establish an hepatitis B virus peptide library by screening high affinity peptides in the high frequency HLA population in China and Northeast Asia, and can also study the related tumorigenesis mechanism by comparing the changes in the antigen peptide and HLA association caused by mutations. In our study, 352 affinity values were predicted from 45 HBV peptides, and the best possible limiting HLA molecules were screened out based on the data (HLA-A * 24:02, A * 02:01, A * 11:01, A * 33:03, A * 02:07), which saved time and costs for subsequent research. Moreover, our prediction was not limited to the known structure of HLA molecules, as we also predicted the HBV limitation of HLA molecules of unknown structure based on homologous modeling and protein docking. At present, a set of structure-based research procedures explored in this research has achieved some preliminary results, but it is still worth improving on. Further studies will be conducted on structure-based affinity prediction to provide additional useful information about peptide–HLA interactions and to improve the prediction time and accuracy. It is believed that in the future, the structure-based prediction method will complement the sequence-based prediction method, saving on labor in the identification of immunogenic peptides, and speeding up the development of tumor vaccines.

## 4. Materials and Methods

### 4.1. HLA Structure Source

In this paper, structures of HLA-A genotypes in the PDB (Protein Data Bank) database were collected by means of MMseqs2 to ensure conformational integrity. As for HLA genotypes that do not have an established structure, such as HLA-A * 33:03, we modelled the structure by homologous modeling and molecular docking. First, we collected the complete HLA I class alpha chain amino acid sequences from the IPD-IMGT/HLA database (http://www.ebi.ac.uk/ipd/imgt/hla/, accessed on 13 April 2020). Second, we used the Advanced model function of Schrödinger software (2020-1, LLC, New York, NY, USA) to construct the HLA alpha chain based on sequence similarity calculated by BLAST. Third, structures were submitted to SAVES (https://saves.mbi.ucla.edu/, accessed on 13 July 2020) for a reliability test, then we fixed the unreasonable conformation and recorded the parameters of each structure. Forth, the refined HLA class I alpha chains and the beta chains from templates were docked by means of molecular docking, accomplished using the CoDockPP online tool [21]. Finally, we evaluated the rationality of the heterodimer in Molprobity [28]. All test parameters of the modelled structures are provided in the Appendix A (Appendix A).

### 4.2. HBV Positive Peptide

The amino acid sequences of four HBV proteins (HBsAg, HBeAg, HBpol and HBx) of four genotypes A, B, C and D were obtained by referring to the Uniprot database (https://www.uniprot.org/, accessed on 13 October 2018). Furthermore, we used six epitope prediction tools (SVMHC-SYFPEITHI/MHCPEP, IEDB-ANN/SMM, NetMHC, SYFPEITHI, BIMAS, EPIJEN) to obtain high frequency HLA-restricted positive epitopes with 9 or 10 amino acids, and eventually 45 HBV-positive peptides for subsequent simulated prediction. Detailed information about these 45 epitopes is presented in Table 6.

### 4.3. Selection of HLA Structure

The majority of the HLA molecules with different genotypes contain a variety of different structures, which are mainly diverse from the peptide segment bound by their antigen binding groove. For the factor in which the binding groove of HLA-I molecules is closed at both ends, antigenic peptides are limited in length and usually consist of 8–11 amino acids. Unlike HLA I molecules, the antigen-peptide binding slots of class II molecules are open, and can therefore accommodate longer peptides, of typically 13 to 18 amino acids. However, the binding of antigen peptides to HLA molecules is inclusive and different HLA molecules can bind to various antigen peptides. Relevant studies have demonstrated that there are 6 binding pockets (A–F pockets) on HLA molecules, but each pocket has a different degree of contribution. In general, the binding force is mainly provided by the hydrogen bond network formed between the second and ninth anchor residues of the antigenic peptide and the B and F bags of the HLA molecule [29]. Therefore, we first performed a sequence analysis on the peptides bound in the collected HLA structures.

HLA complex structures with the same anchor-binding residues as HBV peptides were selected as candidates. For those HLA complexes whose anchors were not consistent with HBV positive peptides, we selected the structures most similar to HBV mutated peptides according to the length, amino acid sequence and crystal resolution of the p-HLA complex for amino acid scanning mutations.

For the groups that it was not suitable to obtain the docking conformation of HBV peptide and HLA by amino acid scanning mutation, we chose to use the protein docking method to obtain reasonable conformation. Ten optimal structures can be obtained based on a knowledge-based scoring function and site constraint protein docking method in CoDockPP [21], in which a distance-based scoring method is trained according to the iterative method of statistical mechanics [22], and the formula is as follows:Score = Σijuij(r)uij stands for effective pairwise potentials

In this article, a total of four structures, A1102, A2601, A3303 and A3101, respectively, are modelled, and this method was adopted for all of them.

### 4.4. Scanning Amino Acid Mutation

The amino acid scanning mutation developed based on the calculation method of binding free energy can change the properties of some residues without transforming the conformation of the protein backbone, so as to explore the influence of the protein function. We use the residue scanning calculation module of BioLuminate (version 1.0, Schrödinger, 2020-1, LLC, New York, NY, USA) to calculate the binding capability of HLA molecules and positive peptides. This module uses the MM-GBSA method, combines the OPLS2005 force field and the VSGB solvent model. Difference in net binding free energy between the wild-type protein and mutant-type protein was calculated using the thermodynamic cycle so as to compare the system energy of the two. Firstly, we selected affinity and stability as the criteria to estimate the structure, manually design and modify candidate mutant residues and set them as simultaneous mutations. After that, we performed sorting in accordance with the protein affinity after mutation, and defined the affinity threshold predicted by MM-GBSA with 3 kal/mol [30]. Due to the fact that HBV positive peptides belong to a minor number of peptides, the scanning of amino acid mutations can achieve site-directed mutation at the binding position of the original antigen peptide of the HLA molecule, effectively eliminating the confusion caused by an inaccurate docking posture. If the number of amino acids mutated at the same time reaches more than 8, it needs to be calculated by script. We used WinSCP software (version 5.17.7) to modify the amino acids. WinSCP securely copies and edits files between local and remote devices. By running the newly modified file script through Xshell we were able to obtain the mutation results.

### 4.5. Molecular Docking

Molecular docking is an effective method for predicting binding patterns and the affinity between ligand receptors and proteins, providing a series of possible theoretical conformations after recognition, and further optimizing the spatial structure of proteins through subsequent molecular dynamics simulations [31]. According to the changing degree of protein conformation, it is mainly divided into the following three categories: rigid docking, semi-flexible docking and flexible docking. The antigenic peptides of HLA molecules are diverse, and part of the peptide chains contain cysteine residues, which may be skipped in the mutant design due to the tendency to form disulfide. Therefore, in order to enhance the rationality of the experiment, we performed a dynamic simulation on the structure that cannot be mutated in residue scanning.

First of all, the molecular docking described in our test uses the Peptide docking of the Glide module, which regards receptors as rigid structures, but can subtly modify the docking of receptors and ligands by softening the active site area. Taking the peptide ligand as the constraint object, we automatically set up the receptor docking box to generate and input the sequence of the positive peptide to be mutated in the Text. Then, we obtained a set of peptide-docking poses. We used the superimposition function of Schrödinger to calculate the conformational similarity between the docking pose and binding pose of the peptide and selected the pose with the lowest RMSD Score.

### 4.6. Molecular Dynamics Simulation

With the evolution of computer technology, mechanism of experimental phenomena can be visually demonstrated through software simulation, which can effectively predict the wet experiment beforehand. The accuracy of the simulation will improve the economic effectiveness of the experiment and make a greater contribution to scientific research. Molecular dynamics simulates the Newtonian dynamics of the model system, which can capture the conformational changes of the protein in the course of the interaction, as well as the coordinates, velocity and energy trajectories of the particles [32]. Schrödinger’s Desmond module is our preferred method for performing high-performance molecular dynamics simulations and supporting GPU acceleration. The docking structure was placed in the prepared simulation system which used an OPLS3E force field and SPC water molecule model. This force field has a higher accuracy for the structural performance and conformation evaluation of the protein ligand [33]. In addition, the overall model system will run a 100 ns molecular dynamics simulation at a temperature of 300K and a pressure of 1bar. According to the laws of thermodynamics, the model system will be prone to a state with the lowest binding free energy and the most stable structure.

### 4.7. Protein Ligand Interaction Analysis

A Simulation Interaction Diagram panel can be applied to visualize the information of the simulated protein–ligand interactions and obtain the RMSD and RMSF values of the protein–ligand as well as the hydrogen bond and hydrophobicity. RMSD can be used for determining whether the protein complex is stable during the simulation process [34]. The molecular systems simulated by molecular dynamics are valuable only when the conformation is extracted and analyzed at equilibrium.

In cases that the RMSD value still displays a fluctuation over a wide range, the simulation architecture changes and requires a longer simulation time. Root Mean Squared Deviation (RMSD) is a distance function and can be calculated to effectively determine whether the structure is in a steady state, which is of great significance for further analysis. The calculation formula is as follows:$$RMSD_{x}=\sqrt{\frac{1}{N}\overset{N}{\underset{i=1}{\sum }}{\left(r_{i}^{\prime }\left(t_{x}\right)-r_{i}\left(t_{ref}\right)\right)}^{2}}$$

In this formula, *N* is the number of atoms selected, and tref is the reference time (generally, the corresponding time of the first frame is *t* = 0). *r*′ is the position of the selected atoms in frame *x* after superimposing on the reference frame, where frame *x* is recorded at time *t_x_*. The procedure is repeated for each frame in the simulation trajectory.

### 4.8. Binding Free Energy Calculation

The theoretical study of protein–ligand interactions and the rapid development of computer-aided medicine design have led to great progress in the calculation of binding free energy. Molecular dynamics simulation indirectly calculates the binding free energy between proteins by establishing a Thermodynamic cycle [35]. It is well known that Free Energy Perturbation (FEN) and Thermodynamic Integration (Ti) are classic methods. However, these two methods have some limitations due to the need to strictly define the computing system and because they are time-consuming [36]. In recent years, the MM/PB (GB) SA method for the combined free energy calculation of molecular dynamics trajectory files has attracted increasing attention. Studies have highlighted that MM/GBSA presents promising performance in predicting the relative binding free energy of protein ligands, and combines precision and speed comprehensively [37].

The calculation formula is as follows: MM/GBSA dG Bind (NS) = Complex-Receptor-Ligand.

The NS = NO Strain, which is means the conformational change of the receptor ligand during the formation of the complex is not considered.

## Figures and Tables

**Figure 1 ijms-23-04629-f001:**
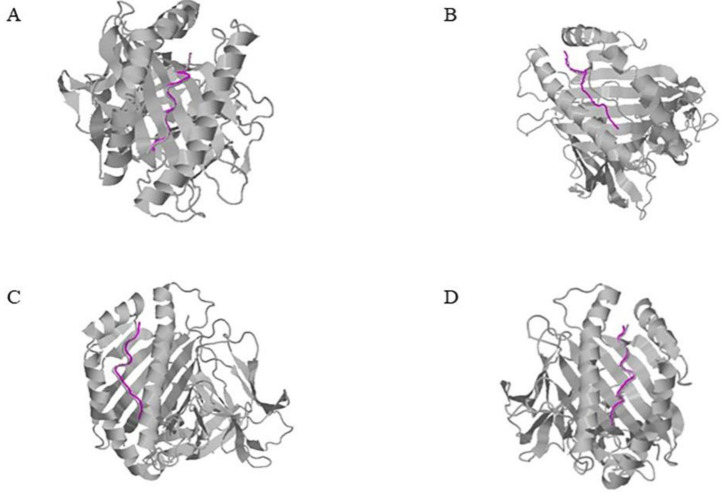
Modelled structure docking model. Ligands for modelled receptor structures and modelled template structures are shown in gray and predictive ligands are colored in magenta. (**A**) HLA-A * 11:02 (**B**) HLA-A * 26:01 (**C**) HLA-A * 31:01 (**D**) HLA-A * 33:03.

**Figure 2 ijms-23-04629-f002:**
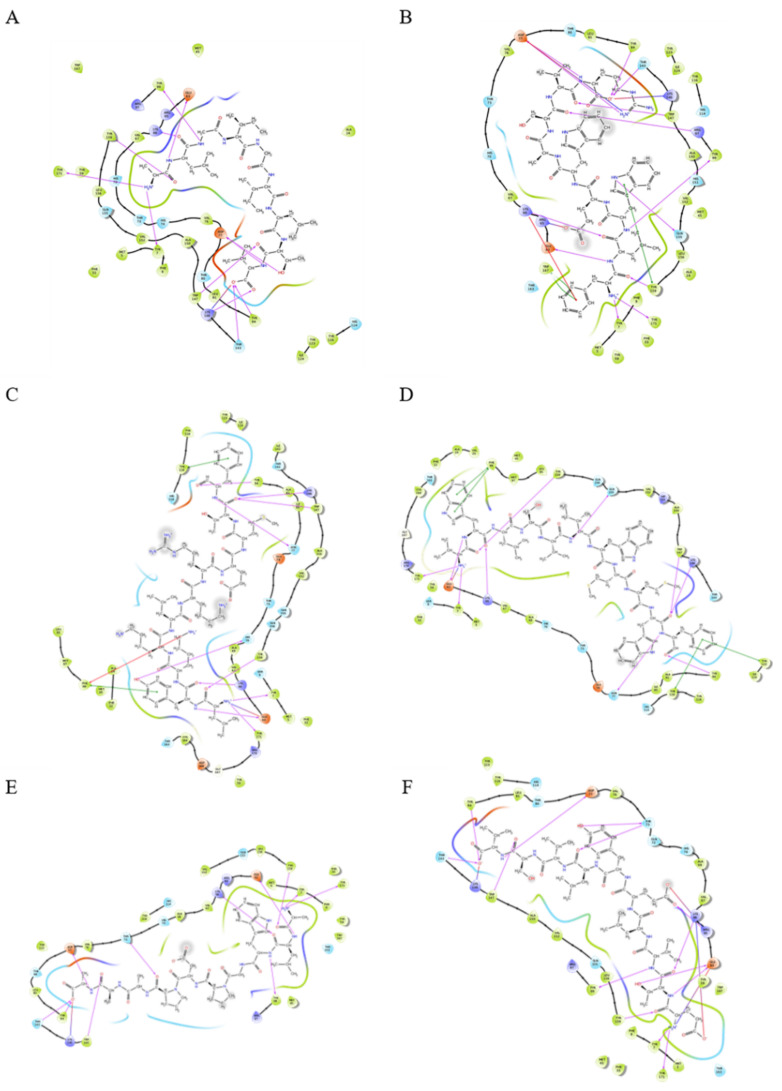
A 2D structure of the interaction between peptide and HLA structure. Residues are represented by the shape of water droplets. The interactions between residues and ligands are represented by lines and colored according to different interaction types. The hydrogen bond is colored in magenta and the Pi-Pi bond is colored in green. Magenta gradient lines represent salt bridge. (**A**) Interaction of HLA-A * 02:01 (PDB ID: 1JHT) and its peptide (ALGIGILTV). Hydrogen bonds are formed between peptide and amino acids TYR7, GLU63, LYS66, ASP77, TYR84, TYR99, THR143, LYS146, TRP147, TYR159 and TYR171. A salt bridge is formed with residue LYS146. (**B**) Interaction of HLA-A * 02:01 (PDB ID: 1JHT) and HBV-positive peptide (FLWEWASVR). Hydrogen bonds are formed between peptide and amino acids TYR7, GLU63, LYS66, ASP77, TYR84, ARG97, TYR99, THR143, TRP147, GLU155, TYR159 and TYR171, Pi-Pi bond is formed with residue TYR159 and salt bridges are formed with residue ARG65, ASP77 and LYS146. (**C**) Interaction of HLA-A * 24:02 (PDB ID: 5XWD) and its peptide (LYKKLKREMTF). Hydrogen bonds are formed between peptide and amino acids TYR7, GLU63, LYS66, HIE70, ASN77, TYR84, LYS146, TRP147, TYR159 and TYR171, Pi-Pi bonds are formed with residue PHE99 and TYR116, salt bridge is formed with residue GLU63. (**D**) Interaction of HLA-A * 24:02 (PDB ID: 5XWD) and HBV positive peptide (VWLSVIWMMW). Hydrogen bonds are formed between peptide and amino acids TYR7, GLU63, LYS66, ASN77, TYR84, LYS146, TRP147, GLU155, TYR159 and TYR171, Pi-Pi bonds are formed with residue PHE99, TYR116 and TYR123, salt bridge is formed with residue GLU63. (**E**) Interaction of HLA-A * 02:01 (PDB ID: 3UTQ) and its peptide (ALWGPDPAAA). Hydrogen bonds are formed between peptide and amino acids TYR7, GLU63, LYS66, THR73, ASP77, TYR99, THR143, LYS146, TRP147, TYR159 and TYR171, salt bridge is formed with residue LYS146. (**F**) Interaction of HLA-A * 02:01 (PDB ID: 3UTQ) and HBV positive peptide (ETVLEYLVSV). Hydrogen bonds are formed between peptide and amino acids TYR7, GLU63, ARG65, LYS66, THR73, ASP77, TYR84, TYR99, THR143, TRP147, TYR159 and TYR171, and salt bridges are formed with residue GLU63, ARG65, LYS66 and LYS146.

**Figure 3 ijms-23-04629-f003:**
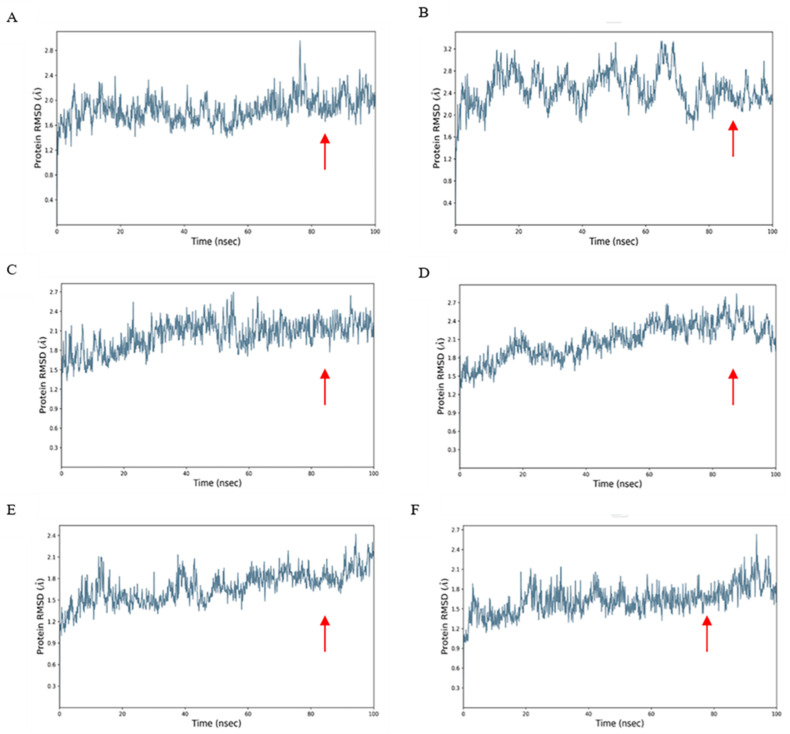
RMSD of the docking structure. The arrow (red) indicates the selected stable regions, where the value fluctuates slightly and can be used for subsequent calculations of the binding free energy. (**A**) The complex 5GRD-SMYPSCCCTK remained stable between 10 and 100 ns. (**B**) The complex 1I4F-SMYPSCCCTK remained stable from 80 to 100 ns. (**C**) The complex 3RL2-SMYPSCCCTK remained stable between 60 and 100 ns. (**D**) Compound A * 26:01-ETVLEYLVSV remained stable from 60 to 100 ns. (**E**) Compound 5GRD-ETVLEYLVSV remained stable from 60 to 100 ns. (**F**) Compound 3UTQ-ETVLEYLVSV remained stable from 20 to 100 ns.

**Table 1 ijms-23-04629-t001:** Selection of HBV-positive peptide and HLA-A genotype.

Positive Peptide Sequence	HLA-A Genotype	PDB-ID	Peptide	Resolution (Å)
FLWEWASVR	A * 02:01	1AO7	LLFGYPVYV	2.6
		1BD2	LLFGYPVYV	2.5
		1DUZ	LLFGYPVYV	1.8
		1QEW	FLWGPRALV	2.2
		1JHT	ALGIGILTV	2.15
		3I6G	GLMWLSYFV	2.2
		5ENW	GLKEGIPAL	1.85
		5F9J	YLSPIASPL	2.51
		5FA3	GLLPELPAV	1.86
	A * 24:02	3I6L	QFKDNVILL	2.4
		2BCK	VYGFVRACL	2.8

The red letters mean the P2 and PΩ residues of antigenic peptides.

**Table 2 ijms-23-04629-t002:** CoDockPP butt mold structure.

HLA-A Genotype	Score Value kal/mol	Ligand RMSD (Å)
A * 11:02	−250.13	0.389
A * 26:01	−332.26	3.100
A * 31:01	−270.40	0.781
A * 33:03	−262.92	0.596

**Table 3 ijms-23-04629-t003:** Part of the HLA mutation performance.

HBV Peptide	HLA-A Genotype	PDB-ID	HLA Peptide	△Affinity kcal/mol	△Stability (Solvated) kcal/mol	△Hydropathy	△Prime Energy kcal/mol	Prime Affinity kcal/mol
FLWEWASVR	A * 02:01	1AO7	LLFGYPVYV	59.73	48.23	0.47	6.36	−231.538
		1BD2	LLFGYPVYV	100.45	57.18	0.31	56.03	−186.495
		1DUZ	LLFGYPVYV	−1	25.04	1.72	−77.56	−215.093
		1QEW	FLWGPRALV	29.19	3.38	0.76	−44.56	−187.811
		1JHT	ALGIGILTV	−25.79	58.4	−1.97	−64.88	−220.932
		3I6G	GLMWLSYFV	101.23	93.87	−1.41	135.48	−116.632
		5ENW	GLKEGIPAL	207.63	162.23	0.49	286.45	−38.59
		5F9J	YLSPIASPL	−10.79	19.31	−0.59	−127.65	−245.795
		6NCA	YVLDHLIVV	−15.94	18.86	0.1	−54.54	−209.7
		5FA3	GLLPELPAV	111.81	62.17	1.55	51.95	−112.188
	A * 33:03	5WJL template	GTSGSPIVNR	−19.43	21.53	−0.70	−56.20	−244.411
	A * 24:02	3I6L	QFKDNVILL	11.43	13.94	0.82	57.69	−195.086
		2BCK	VYGFVRACL	23.14	−4.06	1.14	−24.61	−189.799
VWLSVIWMMW	A * 02:01	5F7D	GLKEGIPALD	98.89	128.14	2.77	241.54	−129.266
		5D9S	FVLELEPEWTV	109.48	101.41	4.06	244.59	−132.558
		5EOT	GLLPELPAVGG	91.86	129.52	2.76	161.08	−141.106
		3I6K	TLACFVLAAV	39.08	41.57	-0.07	56.11	−160.656
		1I4F	GVYDGREHTV	135.81	153.98	6.16	391.17	−65.913
	A * 02:07	3OXS	FLPSDFFPSV	96.17	49.52	4.83	103.16	−142.360
	A * 24:02	5WWI	LYKKLKREMTF	6.77	3.73	6.62	90.22	−240.244
		5WXD	LYKKLKREMTF	−9.87	12.91	7.73	82.75	−243.243
ETVLEYLVSV	A * 26:01	6AT9 template	AQDIYRASYY	76.53	−35.56	2.22	131.59	−109.727
	A * 11:01	5WJL	GTSGSPIVNR	86.11	38.35	0.58	121.02	−125.356
		5WJN	GTSGSPIINR	70.9	30.3	1.15	87.87	−164.241
		5WKF	GTSGSPIVNR	372.07	168.26	0.43	536.89	115.8
		5WKH	GTSGSPIINR	57.75	25.51	2.3	69.94	−165.814
		1QVO	QVPLRPMTYK	55.37	−3.89	0.53	34.32	−164.671
	A * 02:01	5YXU	KLVALGINAV	109.1	66.08	0.18	162.51	−159.462
		1I4F	GVYDGREHTV	179.02	34.6	4.12	284.63	−22.858
		3I6K	TLACFVLAAV	3.67	−4.35	−2.31	−55.59	−194.707
		3UTQ	ALWGPDPAAA	−15.25	27.51	1.61	−41.95	−211.162

**Table 4 ijms-23-04629-t004:** Number of different types of interactions.

HLA-Peptide	Number of Hydrogen Bonds	Number of Salt Bridge	Number of Pi-Pi Bonds
A * 02:01-1JHT- ALGIGILTV	12	1	0
A * 02:01-1JHT-FLWEWAFVR	13	3	1
A * 24:02-5WXD-LYKKLKREMTF	10	1	2
A * 24:02-5WXD-VWLSVIWMMW	10	1	4
A * 02:01-3UTQ-ALWGPDPAAA	11	1	0
A * 02:01-3UTQ-ETVLEYLVSV	14	4	0

**Table 5 ijms-23-04629-t005:** HBV-HLA combined free energy calculation.

HBV Peptide	HLA-A Genotype	PDB-ID	HLA Peptide	Binding Free Energy kcal/mol
SMYPSCCCTK	A * 11:01	5GRD	SSCSSCPLSK	−64.234
	A * 02:01	1I4F	GVYDGREHTV	−80.997
	A * 03:01	3RL2	QVPLRPMTYK	−76.071
ETVLEYLVSV	A * 26:01	modeling		−90.805
	A * 11:01	5GRD	SSCSSCPLSK	−93.877
	A * 02:01	3UTQ	ALWGPDPAAA	−76.755

**Table 6 ijms-23-04629-t006:** Binding analysis of 45 HBV epitopes and HLA-A alleles.

Epitopes	Origin Protein	HBV Genotype(s)	Position (Start–End)	Bioinformatics Prediction
FLPSDFFPSI	HBeAg	B/C	47–56	A * 02:07 > A * 02:01 > A * 11:01
LLDTASALY	HBeAg	A/B/D	59–67	A * 01:01 > A * 11:02
ILCWGELMNL	HBeAg	B/C	88–97	A * 02:07 > A * 24:02
ASRELVVSY	HBeAg	B/C	109–117	A * 02:01 > A * 30:01 > A * 02:06
SYVNVNMGL	HBeAg	A/C/D	116–124	A * 24:02 > A * 02:07
WFHISCLTF	HBeAg	A/B/C/D	131–139	A * 24:02 > A * 02:07 > A * 02:01
ETVLEYLVSV	HBeAg	C	142–151	A * 02:01 > A * 11:01 > A * 26:01
ILSTLPETTV	HBeAg	A/B/C/D	168–177	A * 02:01 > A * 02:03
STLPETTVVR	HBeAg	A/B/C/D	170–179	A * 11:01 > A * 02:07
ASPISSIFSR	HBsAg	C	158–167	A * 11:01
SPISSIFSR	HBsAg	C	159–167	A * 02:01 > A * 11:01
SAISSISSK	HBsAg	B	159–167	A * 11:01>A * 02:03
LQAGFFSLTK	HBsAg	B	189–198	A * 02:07 > A * 24:02 > A * 11:02
QAGFFSLTK	HBsAg	B	190–198	A * 11:01 > A * 31:01
QAGFFLLTR	HBsAg	C/D	190–198	A * 11:01 > A * 11:02
CPGYRWMCLR	HBsAg	A/B/C/D	243–252	A * 33:03
LFILLLCLI	HBsAg	A/C/D	258–266	A * 24:02
LLDYQGMLPV	HBsAg	A/B/C/D	271–280	A * 02:03 > A * 24:02
SMYPSCCCTK	HBsAg	C	306–315	A * 02:01 > A * 11:01 > A * 03:01
FLWEWASVR	HBsAg	C	335–343	A * 02:01 > A * 33:03 > A * 24:02
VWLSVIWMMW	HBsAg	A/B/C/D	364–373	A * 24:02 > A * 02:01 > A * 02:07
MMWYWGPSL	HBsAg	A/C/D	371–379	A * 02:01 > A * 02:07 > A * 24:02
MMWFWGPSL	HBsAg	B	371–379	A * 24:02 > A * 02:07 > A * 03:01
MMWYWGPSLY	HBsAg	A/C/D	371–380	A * 24:02 > A * 03:01 > A * 02:07
LYSILSPFL	HBsAg	C/D	379–387	A * 24:02 > A * 02:01
TVNAHQILPK	HBx	A/D	82–91	A * 11:01 > A * 30:01
TVNAHGNLPK	HBx	B	82–91	A * 02:01 > A * 11:01 > A * 03:01
TVNAHQVLPK	HBx	C	82–91	A * 11:01
STTDLEAYFK	HBx	A/B/C/D	104–113	A * 02:01 > A * 11:01 > A * 02:06
RLKVFVLGG	HBx	A/B/C/D	128–136	A * 24:02 > A * 30:01
KVFVLGGCR	HBx	A/B/C/D	130–138	A * 24:02 > A * 31:01
VCAPAPCNF	HBx	D	142–150	A * 24:02
LYSSTVPCF	HBpol	B	62–70	A * 24:02 > A * 33:03 > A * 02:01
LYSSTVPVF	HBpol	C	62–70	A * 24:02 > A * 02:07 > A * 02:01
FYPKVTKYL	HBpol	D	115–123	A * 24:02 > A * 11:01
KVTKYLPLDK	HBpol	D	118–127	A * 11:01
TLWKAGILYK	HBpol	A/B/C/D	150–159	A * 03:01 > A * 24:02 > A * 11:02
FLLAQFTSA	HBpol	A/B/C/D	524–532	A * 02:03 > A * 02:07
LLAQFTSAI	HBpol	A/B/C/D	525–533	A * 02:03
PTYKAFLCK	HBpol	C/D	671–679	A * 11:01 > A * 02:06
HTAELLAACF	HBpol	A/B/C/D	726–735	A * 24:02 > A * 26:01
RSRSGAKLI	HBpol	B/C	737–745	A * 02:01 > A * 02:06 > A * 30:01
RSRSGANIL	HBpol	B/C	737–745	A * 02:01 > A * 30:01
KLIGTDNSV	HBpol	A/C	743–751	A * 02:01 > A * 02:03
KLIGTHNSV	HBpol	B	743–751	A * 02:01 > A * 02:03 > A * 11:01

## Data Availability

The data presented in this study are available in Ding Y., Zhou Z., Li X., Zhao C., Jin X., Liu X., Wu Y., Mei X., Li J., Qiu J. and Shen C. (2022) Screening and Identification of HBV Epitopes Restricted by Multiple Prevalent HLA-A Allotypes. Front. Immunol. 13:847105, doi:10.3389/fimmu.2022.847105.

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
