# Peer review of "The Use of Molecular Dynamics Simulation Method to Quantitatively Evaluate the Affinity between HBV Antigen T Cell Epitope Peptides and HLA-A Molecules"

_ijms, 2022, doi:10.3390/ijms23094629_

Round 1

Reviewer 1 Report

The authors provide a new method for predicting HBV antigen peptides restricted to the HLA-A allele by evaluating the binding affinity of HBV antigen peptides to HLA molecules. This study will contribute meaningful information and future therapeutic strategies for the mechanism of disease pathogenesis related to HBV infection by the combination of HLA and HBV peptides. However, a more detailed discussion of the causal relationship with previous reports is requested

  • Please provide detailed information (amino acid position, origin, etc.) on the 45 HBV-positive peptides selected from the 4 HBV proteins.
  • There have been many reports on the relationship between HBV infection and HLA class I alleles. How consistent are the results of your analysis based on the affinities between HBV-positive peptides and HLA-A found by the authors and others?
  • According to the authors' report, a reasonable docking conformation of HBV peptides and HLA molecules, i.e., a combination with high affinity between the two molecules, can induce killer T cells, which is assumed to mean that it can protect against HBV infection. Considering this, it may be more effective to examine the risk factors that promote the development of infection (HLA class I alleles in combination with HBV peptides) to prevent HBV infection.
  • The expression of line 46-47 in the Introduction is insufficient; it should be changed to HLA class I gene can be divided into two categories classical class I(HLA-A,-B,-C), which are highly genetically polymorphic, and non-classical I (HLA-E,-F,-G). Classical Class I genes are responsible for delivering the peptide (typically 8-12 amino acids in length) to CD8+ T cells.

Author Response

Response to Reviewer 1 Comments

Dear Editors and Reviewers:

Thank you for giving us an opportunity to revise our manuscript. We appreciate for the reviewers’ valuable comments for our manuscript entitled “The use of molecular dynamics simulation method to quantitatively evaluate the affinity between HBV antigen T cell epitope peptides and HLA-A molecules” (ID: ijms-1645274). We have revised our manuscript according to the comments. Each comment has been well addressed and the related revision has been accurately incorporated. The revised portions are highlighted in yellow in the manuscript.

We hope that the revised manuscript can be qualified for publishing in International Journal of Molecular Sciences.

Thank you very much!

Best regards,

Yours sincerely,

Jian Li

Comments and Suggestions for Authors

The authors provide a new method for predicting HBV antigen peptides restricted to the HLA-A allele by evaluating the binding affinity of HBV antigen peptides to HLA molecules. This study will contribute meaningful information and future therapeutic strategies for the mechanism of disease pathogenesis related to HBV infection by the combination of HLA and HBV peptides. However, a more detailed discussion of the causal relationship with previous reports is requested.

Response: Thank you very much for your constructive comments. We have added some previous reports on the link between HLA and HBV to the manuscript and found that there are several strands of evidence for an important role of the CD8+ T cell response in HBV. Associations have been documented between HLA-A and disease outcome, including responses to, and recovery from, acute infection. There are also some databases of HLA class I epitopes in hepatitis B virus, such as Hepitopes, but they are mainly limited by HLA-A*02:01, A*24:02 or B*07:02, which are common supertypes in Caucasians. Therefore, we hope to predict the binding affinity between HLA molecules and HBV antigenic peptides by studying the effects of mutations on binding affinity and stability, and reasonably screen out the restrictive antigenic peptides targeting high frequency HLA-A in Chinese and Northeast Asian populations. Based on our research process, we can carry out quantitative sequencing according to the affinity value between antigen peptide and HLA, establish hepatitis B virus peptide library by screening high affinity peptide for the high frequency HLA population in China and Northeast Asia, and can also study the related tumorigenesis mechanism by comparing the changes of antigen peptide and HLA association caused by mutations. (Lines 54-66 of page 2) (Lines 320-331 of page 11)

Point 1: Please provide detailed information (amino acid position, origin, etc.) on the 45 HBV-positive peptides selected from the 4 HBV proteins.

Response 1: We are grateful to reviewer for this suggestion. We have rearranged the statistics and added the information of the 45 HBV-positive peptides (Table 6) in the manuscript. (Lines 257-258 of page 9)

Point 2: There have been many reports on the relationship between HBV infection and HLA class I alleles. How consistent are the results of your analysis based on the affinities between HBV-positive peptides and HLA-A found by the authors and others?

Response 2: Thank you for your rigorous consideration. Anthony et al. have confirmed a strong interaction between epitope FLPSDFFPSI and HLA-A* 02:01, which is consistent with our prediction. For all the epitopes we predicted, we carried out experimental verification using cell lines, there was 68% consistence between our experimental verification and the bioinformatic prediction. Specific descriptions have been added in the manuscript (Lines 241-255 of page 8) (Lines 320-331 of page 11)

Point 3: According to the authors' report, a reasonable docking conformation of HBV peptides and HLA molecules, i.e., a combination with high affinity between the two molecules, can induce killer T cells, which is assumed to mean that it can protect against HBV infection. Considering this, it may be more effective to examine the risk factors that promote the development of infection (HLA class I alleles in combination with HBV peptides) to prevent HBV infection.

Response 3: We are grateful for the suggestion. HBV is a non-cytopathic virus, and the degree of liver damage in chronic HBV infection is driven by the activation of the immune system. The TCR molecule recognizes and binds to HLA-peptide complexes, triggering a cascade of immune responses. In this process, most computer predictions have focused primarily on affinity between HLA and peptide. We look for the restrictive HLA of HBV epitopes from molecular structural, and of course we will further explore the major factors in T cell reception of activation signals.

Point 4: The expression of line 46-47 in the Introduction is insufficient; it should be changed to HLA class I gene can be divided into two categories classical class I (HLA-A, -B, -C), which are highly genetically polymorphic, and non-classical I (HLA-E, -F, -G). Classical Class I genes are responsible for delivering the peptide (typically 8-12 amino acids in length) to CD8+ T cells.

Response 4: We are sorry for our incorrect writing. Thank you very much for your instructive suggestion. We have revised it in the manuscript. (Lines 47-51 of page 2)

Reviewer 2 Report

The authors explained a new method in predicting the affinity between HBV antigen T cell epitode peptides and HLA-A alleles.

Major comments

  1. The methods proposed are worth encouraging but contained severe bias, especially in the selection of HLA genes to predict binding efficacy. HLA class II are known to be the major HLA genes to be associated with HBV, however, the authors have chosen HLA-A for the binding prediction. The paper did not state the reason why HLA-A was chosen. It is understandable that HLA-A is a good start due to its close conformation compared to HLA class II genes, the authors can choose another disease known to be strongly associated with HLA class I instead of using HBV as a model.

  1. HLA-A genes that are absent from the PDB database are predicted using homology modeling and molecular docking. The reviewer suggests using alphafold2 for the prediction of the structure instead of using the method mentioned above as alphafold2 are proven to be highly accurate.

  1. Please kindly show the differences between the binding affinity predicted by other software such as NetMHCpan and the methods that the authors suggested.

  1. The paper would need to be supported some functional study to convince the reader of the accuracy of the binding prediction.

  1. The conclusion of the paper is unclear in the way that which HLA-A alleles are known to have the highest possibility of binding to HBV peptides are hard to understand

Author Response

Response to Reviewer 2 Comments

Dear Editors and Reviewers:

Thank you for giving us an opportunity to revise our manuscript. We appreciate for the reviewers’ valuable comments for our manuscript entitled “The use of molecular dynamics simulation method to quantitatively evaluate the affinity between HBV antigen T cell epitope peptides and HLA-A molecules” (ID: ijms-1645274). We have revised our manuscript according to the comments. Each comment has been well addressed and the related revision has been accurately incorporated. The revised portions are highlighted in yellow in the manuscript.

We hope that the revised manuscript can be qualified for publishing in International Journal of Molecular Sciences.

Thank you very much!

Best regards,

Yours sincerely,

Jian Li

Comments and Suggestions for Authors

The authors explained a new method in predicting the affinity between HBV antigen T cell epitode peptides and HLA-A alleles.

Major comments

Point 1: The methods proposed are worth encouraging but contained severe bias, especially in the selection of HLA genes to predict binding efficacy. HLA class II are known to be the major HLA genes to be associated with HBV, however, the authors have chosen HLA-A for the binding prediction. The paper did not state the reason why HLA-A was chosen. It is understandable that HLA-A is a good start due to its close conformation compared to HLA class II genes, the authors can choose another disease known to be strongly associated with HLA class I instead of using HBV as a model.

Response 1: Thank you for your valuable advice. HLA-A genes are closely related to immune response pathways, and the diversity of HLA-A is a protective barrier against bacterial and viral invasion. In our study, we wonder the association between HBV infection and T-cell immune responses in Chinese and Northeast Asian populations, so we selected 13 HLA-A genotypes with a total frequency of approximately 95% in these populations from IPD-IMGT/HLA database. We totally understand the reviewer’s concern and based on the previous version, we address the content in the section of introduction according to your thoughtful comments. This is just the start of the end, we shall continue our work to deeply understand the HBV disease. (Lines 54-66 of page 2)

Point 2: HLA-A genes that are absent from the PDB database are predicted using homology modeling and molecular docking. The reviewer suggests using alphafold2 for the prediction of the structure instead of using the method mentioned above as alphafold2 are proven to be highly accurate.

Response 2: We thank the reviewer for this insightful comment. We agree with the reviewer’s recommendation of Alphafold2 for its high accuracy and the reviewer’s concern is of importance for our further study. However, HLA alleles are highly homologous, and their polymorphism is not evenly distributed throughout the whole molecule, but clustered in the antigen binding groove. Changes of amino acids in several regions of the binding groove will change the shape of the groove, thus changing the peptide binding specificity of HLA molecules. And we have also done a lot of research on the tools used in the manuscript to ensure certain accuracy before the prediction. Strictly speaking, AlphaFold2 is chosen to solve the problem of single-domain protein structure prediction, whereas for HLA molecules, it consists of α and β chains. So instead of using Alphafold2, we constructed the alpha chain of HLA using homologous modeling and heterodimer structures of alpha and beta using CoDockPP. While there is no doubt that we will consider using Alphafold2 in future studies.

Point 3: Please kindly show the differences between the binding affinity predicted by other software such as NetMHCpan and the methods that the authors suggested.

Response 3: Thank you for this valuable comment. We used six epitope prediction tools (SVMHC-SYFPEITHI/MHCPEP, IEDB-ANN/SMM, NetMHC, SYFPEITHI, BIMAS, EPIJEN) to obtain high frequency HLA-restricted positive epitopes with 9 or 10 amino acids and we have classified these methods in the manuscript. As for NetMHCpan, a peptide-MHC prediction method based on machine learning, is similar with NetMHC. Although such tools have established a unified prediction model for all HLA typing, they have a prediction bias for HLA typing with large data volume. In summary, our purpose is to make a comprehensive evaluation, integrating multiple prediction methods will be better than a single method, improve the accuracy of prediction. (Lines 85-90 of page 2)

Point 4: The paper would need to be supported some functional study to convince the reader of the accuracy of the binding prediction.

Response 4: We thank the reviewer for providing this suggestion. In fact, the 45 HBV epitopes were verified as real epitopes by ex vitro enzyme-linked immuno-sorbent spot (ELISPOT) and in vitro co-culture (using patients' peripheral blood mononuclear cells). In addition, we have experimentally verified the binding ability of these peptides to HLA by flow cytometry, and categorized affinity levels to low, medium, and high. The results obtained through our prediction process are 68% consistent with the experiment. We have requoted relevant literatures in the manuscript. At the same time, it also shows the rationality of our forecasting process, our bioinformatic methods can effectively quantitatively predict the situation, and provide specific affinity values that can be compared between different groups, which complements the limitations of the experimental method and is suitable for pre-liminary prediction in the absence of experimental samples or in the case of excessive sample size to provide clues. (Lines 301-305 of page 10)

Point 5: The conclusion of the paper is unclear in the way that which HLA-A alleles are known to have the highest possibility of binding to HBV peptides are hard to understand.

Response 5: We are sorry that the conclusion description of the article is not clear enough. In the results section, 45 HBV epitopes and HLA alleles were combined for being analyzed and highlighted. We found that most of the epitopes have a strong interaction with HLA-A*24:02, HLA-A*02:01 and HLA-A*11:01. Among these 45 epitopes, we identified 19 epitopes that strongly interacted with different HLA alleles and were more likely to be presented to cell surface, which was 68% consistent with our experimental verification. Among them, ten epitopes (ILCWGELMNL, SYVNVNMGL, WFHISCLTF, VWLSVIWMMW, MMWYWGPSL, LYSILSPFL, RLKVFVLGG, LYSSTVPCF, LYSSTVPVF and FYPKVTKYL) have a high binding affinity for HLA-A*24:01, eight epitopes (FLPSDFFPSI, WFHISCLTF, ETVLEYLVSV, ILSTLPETTV, SPISSIFSR, SMYPSCCCTK, FLWEWASVR and MMWYWGPSL) have a high binding affinity to HLA-A*02:01. The epitopes STLPETTVVR and QAGFFLLTR are more likely to bind to HLA-A*11:01, while the epitopes CPGYRWMCLR and FLWE-WASVR have a high binding affinity for HLA-A*33:03, the epitope FLPSDFFPSI has a high ability binding to HLA-A*02:07. (Lines 241-255 of page 8)

Reviewer 3 Report

Good job, very interesting topic, the only thing I don't find clear is the description of the bioinformatics models, I would describe them better in the materials and methods.

Author Response

Response to Reviewer 3 Comments

Dear Editors and Reviewers:

Thank you for giving us an opportunity to revise our manuscript. We appreciate for the reviewers’ valuable comments for our manuscript entitled “The use of molecular dynamics simulation method to quantitatively evaluate the affinity between HBV antigen T cell epitope peptides and HLA-A molecules” (ID: ijms-1645274). We have revised our manuscript according to the comments. Each comment has been well addressed and the related revision has been accurately incorporated. The revised portions are highlighted in yellow in the manuscript.

We hope that the revised manuscript can be qualified for publishing in International Journal of Molecular Sciences.

Thank you very much!

Best regards,

Yours sincerely,

Jian Li

Comments and Suggestions for Authors

Point 1: Good job, very interesting topic, the only thing I don't find clear is the description of the bioinformatics models, I would describe them better in the materials and methods.

Response 1: Thank you very much for your instructive suggestion. In this study, we constructed the structures of four HLA-A genotypes using homologous modeling and molecular docking modeling. The HLA structure source are described in the first part of the manuscript's materials and methods. The specific steps are described below, and we hope to give you a clear explanation. First, we collected the complete HLA І class alpha chain amino acid sequences from IPD-IMGT/HLA database (http://www.ebi.ac.uk/ipd/imgt/hla/). Second, we used the Advanced model function of Schrödinger software (LLC, NY, USA, 2020-1) to construct the HLA alpha chain based on sequence similarity calculated by BLAST. Third, structures were submitted to SAVES (https://saves.mbi.ucla.edu/) for reliability test, we fixed the unreasonable conformation and recorded the parameters of each structure. Forth, refined HLA class І alpha chains and the beta chains from templates were docked by means of molecular docking, accomplished by CoDockPP online tool. Finally, we evaluated the rationality of the heterodimer in Molprobity. (Lines 345-358 of page 11)

Round 2

Reviewer 1 Report

I have no particular comment on the content of the revised article by the authors.

Reviewer 2 Report

The revised manuscript is acceptable